# Advances and Perspectives in Dental Pulp Stem Cell Based Neuroregeneration Therapies

**DOI:** 10.3390/ijms22073546

**Published:** 2021-03-29

**Authors:** Jon Luzuriaga, Yurena Polo, Oier Pastor-Alonso, Beatriz Pardo-Rodríguez, Aitor Larrañaga, Fernando Unda, Jose-Ramon Sarasua, Jose Ramon Pineda, Gaskon Ibarretxe

**Affiliations:** 1Department of Cell Biology and Histology, Faculty of Medicine and Nursing, University of the Basque Country (UPV/EHU), 48940 Leioa, Spain; jon.luzuriaga@gmail.com (J.L.); bea.pardo.r@gmail.com (B.P.-R.); fernando.unda@ehu.eus (F.U.); 2Polimerbio, Paseo Mikeletegi 83, 28009 Donostia-San Sebastián, Spain; ypolo@polimerbio.com; 3Group of Science and Engineering of Polymeric Biomaterials (ZIBIO Group), Department of Mining, Metallurgy Engineering and Materials Science, POLYMAT, University of the Basque Country (UPV/EHU), 48013 Bilbao, Spain; aitor.larranagae@ehu.eus (A.L.); jr.sarasua@ehu.eus (J.-R.S.); 4Department of Neurology, University of California San Francisco, San Francisco, CA 94143, USA; oierpastor90@gmail.com; 5Achucarro Basque Center for Neuroscience Fundazioa, 48940 Leioa, Spain

**Keywords:** dental pulp stem cells, neuroregeneration, neuronal differentiation, neural markers, neuroprotection, immunomodulation, extracellular vesicles, tissue engineering, scaffolds, cell therapy

## Abstract

Human dental pulp stem cells (hDPSCs) are some of the most promising stem cell types for regenerative therapies given their ability to grow in the absence of serum and their realistic possibility to be used in autologous grafts. In this review, we describe the particular advantages of hDPSCs for neuroregenerative cell therapies. We thoroughly discuss the knowledge about their embryonic origin and characteristics of their postnatal niche, as well as the current status of cell culture protocols to maximize their multilineage differentiation potential, highlighting some common issues when assessing neuronal differentiation fates of hDPSCs. We also review the recent progress on neuroprotective and immunomodulatory capacity of hDPSCs and their secreted extracellular vesicles, as well as their combination with scaffold materials to improve their functional integration on the injured central nervous system (CNS) and peripheral nervous system (PNS). Finally, we offer some perspectives on the current and possible future applications of hDPSCs in neuroregenerative cell therapies.

## 1. Introduction: Neural and Mesenchymal Stem Cells, and Neuroregenerative Cell Therapies

Neuroregenerative therapies have always been a priority for health research in developed countries due the overwhelming social, economic and dependency burdens suffered by both the affected patients and their close relatives [1,2,3]. The nervous system in humans possesses a very limited capacity of self-repair in the event of injury. This is the reason why nerve lesions caused by trauma or neurodegenerative diseases often result in highly disabling irreversible conditions and chronic dependency [4,5,6]. Unlike other cells of the body, dead or damaged neurons cannot be easily replaced. Neurogenesis takes place in the developing brain, but it declines in the adulthood. Moreover, neuroinflammation and gliosis following neural trauma or disease make the tissue refractory to the rooting and establishment of new neural connections [7,8,9,10].

Neurogenesis is driven by specific multipotent stem cells known as neural stem cells (NSCs), which give rise to both neurons and glial cells. Embryonic NSCs are the first stone paving the way to brain (re)generation and impairments in their correct function are associated to several types of cortical malformations, with dramatic outcomes on the life of an individual [11]. Although the neurogenic capacity is significantly decreased in the adult mammalian brain, two regions bear NSCs during the entire lifetime of different species. The lateral ventricles and the hippocampus harbor adult NSCs capable of triggering a staggered process that ends up in the integration of a newborn neuron into the adult neuronal circuitry [12,13,14]. In humans, adult newborn neurons have been detected in the lateral ventricles, yet in highly infrequent basis [15,16,17]. In the hippocampus, controversial results over the existence of adult neurogenesis have been recently reported and the topic remains currently under hot debate [16,18]. Regardless, there is agreement about the presence of these neurogenic niches during the first years of life in young infants [16,17,19,20], a time period when NSC dysfunctions might play an important role in the development and chronification of neurodegenerative diseases [21]. Indeed, studies in rodents have shown that adult NSCs undergo changes when facing neurodegenerative challenges, including morphological and functional abnormalities that lead to disruption of neurogenesis and contribute to the detrimental tissue environment in these regions [22,23,24]. The existence of NSCs that could react in pathological conditions has been also suggested in other areas, like the cerebral cortex and spinal cord [25,26]. The amygdala, although with rare adult neurogenic events in normal conditions, has also been postulated to bear quiescent NSCs that could get activated upon peripheric lesions, at least in primates [27].

The scarce numbers of adult NSCs or their aberrant alterations in neurodegenerative diseases suppose the lack of a reliable endogenous mechanism to replenish neurons in the event of their loss. Not in vain, stem cells offer the potential to reduce deleterious signaling and improve traumatic lesions [28,29], and also to slowdown the progression of devastating neurodegenerative diseases such as Huntington’s (HD) [30,31], Parkinson’s (PD) [32,33] or Alzheimer’s disease (AD) [34,35,36]. The idea of using stem cells to treat neurodegenerative diseases was proposed very long ago, obtaining valuable and abundant data using fetal human tissue [31,37,38] or induced pluripotent stem cells (IPSCs) [39,40]. However, these methods raise both safety and ethical concerns that are still under intense debate [41,42,43,44,45,46]. The main practical problems are the security, the very low yields of extraction, and the troublesome conditions of intervention on premature infants to harvest human NSCs [34]. Stem cells from the spinal cord of 8-week fetuses have been tested in clinical trials for chronic spinal cord injury [47,48]. However, it is unlikely that these strategies will ever reach a widespread implementation, due to the scarcity of embryo donors and the associated ethical issues. IPSCs have been proposed to overcome ethical concerns about the use of human embryos. IPSCs can be very efficently differentiated to neurons and glial cells [49,50,51] and they have been proposed as a promising alternative for cell therapy in brain and spinal cord injury [39], as a tool to screen genetic bases of neurological diseases [52] or even as an approach to correct alterations in chronic neurodegenerative diseases [53]. However, a better understanding is still required to regulate the generation of specific neuronal and glial populations in a balanced and coordinated manner. Furthermore, the increased risk of cancer related to the use of IPSCs is regarded as a major drawback for autologous personalized neuroregenerative therapy [49].

In view of the limitations of endogenous NSCs and pluripotent stem cells, it is not surprising that the research community has turned its eyes to alternative sources of stem cells with neural regeneration capacity. Of all of them, the ones that seem the best positioned are mesenchymal stem cells (MSCs), which can give rise to all cell lineages of both proper and specialized connective tissues, including bone, cartilage, muscle and adipose cells, among others. MSCs can be extracted from different sources like the bone marrow, the adipose tissue and the umbilical cord [54]. Human dental pulp stem cells (hDPSCs) had also been traditionally included within MSCs, because they fulfill the standard criteria of plastic-adherent growth, multilineage differentiation and a characteristic molecular marker expression as defined by the presence of CD73, CD90 and CD105, which are required by International Society of Cell Therapy to classify a cell type as a MSC [55], in addition to other accessory markers like CD27, CD29, CD44, CD146, CD166, CD271 and STRO-1 [56,57]. This marker expression profile can be found in hDPSCs, as well as in MSCs from many other tissue sources [58]. On the contrary, MSCs and hDPSCs do not express CD45 (hematopoietic marker), CD14 (monocyte or macrophage marker), CD19 (B cell marker) or MHC-II (major histocompatibility complex II) surface molecules [56,57,59]. MSCs have raised substantial hopes for the clinical management of neural lesions, with very promising results [60]. In this context, hDPSCs have particularly interesting features that encourage their application in neuroregenerative cell therapies even beyond more conventional types of MSCs, as we discuss in the next sections.

## 2. DPSCs as Neural Crest Stem Cells. The postnatal DPSC Niche

Compared to other MSC sources, dental stem cells and DPSCs were discovered relatively late. It was not until the advent of the XXI century that the presence of stem cells in the postnatal human dental pulp was reported [61]. Later on, many other related MSCs with similar characteristics to DPSCs were discovered in other nearby dental tissues, like the periodontal ligament [62], the gingival mucosa [63], the apical papilla [64], the dental follicle [65] or the dental pulp of childhood deciduous teeth [66], among others. Over the last decades hDPSCs have remained as the most extensively studied type of dental stem cells, because of their ease of extraction, absence of ethical issues and relative abundance as biological waste from dental clinics.

It soon became apparent that hDPSCs possessed multilineage differentiation potential that exceeded that of conventional MSCs [59,67,68]. Contrary to other MSC sources, the dental pulp tissue is generated by the neural crest, a structure formed at the fusing borders of the neural tube during development. Cells of the neural crest undergo an epithelial-mesenchymal transition and acquire migratory ability, thus extensively colonizing other parts of the embryo, including the pharyngeal arches which are the precursors of craniomaxillofacial organs and tissues (Figure 1). These neural crest stem cells can subsequently commit to generate the diverse tissues of the oral cavity. Some neural crest stem cells differentiate to MSCs to generate oral neural crest-derived mesenchyme (i.e., ectomesenchyme) which will then give rise to the different oral connective tissues, cartilages, muscles and bones. However, these neural crest stem cells are also the precursors of the cranial peripheral nerve system [69]. Perhaps due to their shared origin, it is not uncommon to observe that hDPSCs express a varied repertoire of both neural progenitor and mature cell markers, even in normal standard (control) culture conditions [59,70,71,72]. Some of the neural markers that are most prominently expressed by hDPSC cultures include Neuroectodermal Stem Cell Intermediate filament marker (Nestin), β-3 tubulin (Tuj1), neurotrophin receptors, and neurofilaments [71,72]. As it can be expected from neural crest-related cells, hDPSC cultures also express neural crest markers like Snail, Slug, Sox10 and HNK1, and also pluripotency-related core factors like Oct4, Sox2 and Nanog [71]. Importantly, the expression of neural crest and pluripotency markers by hDPSCs and the corresponding stemness of these cells can be stimulated by the transient activation of specific signaling pathways, in the absence of any genetic modification [71,73]. Some of these treatments (e.g., Wnt/β-catenin signaling stimulation) have been shown to substantially modify the epigenetic and metabolic footprint of hDPSCs [74,75]. Of particular importance to cell therapy, hDPSCs have a great adaptability to adverse metabolic conditions [76], and can also secrete a large variety of neuroprotective and immunomodulatory factors (discussed in Section 6 of this manuscript) which make them a very attractive tool to promote neural regeneration.

The relationship between DPSCs and nerve tissue goes beyond a shared embryonic origin. The postnatal niches of DPSCs are the neurovascular bundles of the dental pulp, which are intricate associations of peripheral nerves and blood vessels which cross together through the apical foramen to irrigate and innervate the dental pulp tissue (Figure 2). The niche of DPSCs is thus extraordinarily rich in nerve fibers and blood vessels, in contrast to the surrounding loose connective tissue of the rest of the dental pulp [77]. Immunolabeling of STRO-1 expressing cells revealed that hDPSCs were located precisely within these perivascular niches [78]. Later on, lineage tracing experiments in a murine model revealed that stem cells of the dental pulp were neural crest marker-expressing cells associated with neurovascular bundles [79]. Thus, the same DPSC population could ultimately give rise to both non-mesenchymal (e.g., Schwann cells) and mesenchymal (e.g., odontoblasts) lineage-derived cells [79]. Probably because of their close association with neurovascular structures of the dental pulp, hDPSCs also have a very high capacity to generate vascular cells like endothelia and pericytes [80,81]. This higher ability to differentiate to vascular cells comes at the expense of a reduced capacity for commitment to other more conventional types of mesenchymal-related cell lineages, like chondrocyte differentiation [82]. Thus, according to this model, DPSCs would be located at a similar level to neural crest cells within the stem cell hierarchy, with a higher capacity to generate neural and vascular cells than conventional MSCs. Interestingly, DPSCs also exhibit natural niche homing characteristics when they are engrafted in vivo, as they tend to spontaneously migrate to nerves and vascular structures of the host organism after transplantation [80,83].

## 3. Multilineage Differentiation Characteristics and Protocols

As it has been described for other neural crest-derived cells, DPSCs have also a particularly high capacity for multilineage differentiation. Of course, the final differentiation outcome will depend on the signals that DPSC cultures are exposed to. Traditionally, hDPSCs had been regarded as MSCs (or rather, as ectomesenchymal stem cells) because they were shown to differentiate to adipocytes, smooth muscle cells, chondrocytes and osteoblasts, which are all known mesenchymal-derived cell lineages [59,67,83]. This assumption was in no small part sustained by the fact that the manipulation of hDPSC cultures was often made using media containing high proportions (10–20%) of fetal animal serum, which induces the differentiation of DPSCs to MSCs, and eventually to bone/dentin producing cells [84]. Clearly, the maintenance of hDPSC cultures in serum containing media disguised the non-mesenchymal differentiation abilities of these cells. In the presence of fetal bovine serum (FBS), hDPSCs grow in a plastic-adherent mode thus resembling typical MSCs. It was not until the adoption of new serum-free media based culture protocols that the non-mesenchymal differentiation abilities of hDPSCs began to be unveiled. An increasing number of authors began associating the success of neuronal differentiation protocols to the absence of serum, where a neuronal-like differentiation was usually accomplished by the substitution of FBS by diverse differentiation media at the last stages of hDPSC culture [85,86]. However, the presence of 10–20% fetal serum was often maintained for the initial phases of expansion of the parental hDPSC cultures, which compromised the capacity of differentiation of non-mesenchymal lineages at the expense of an increased capacity of differentiation to mesenchymal lineages.

To maximize the neurodifferentiation ability of hDPSCs and eventually consider their application in neuroregenerative therapies, the use of animal serum should be discouraged. Even the transient presence of xenogeneic serum elements during in vitro culture could cause an undesired immune reactivity and even rejection of the transplanted cells [87,88]. The risks of immune rejection are particularly exacerbated in the case of neuroregenerative cell therapies, since the central nervous system is very sensitive to inflammation [89]. Indeed, an inflammatory process due to graft rejection within the brain could easily lead to a brain edema with potentially very dire consequences. Since stem cells were defined as cell-based medicinal products (CBMPs), hDPSCs also need to be prepared under strict culture conditions in order to achieve good manufacturing practice (GMP) required quality standards, as described in EU Regulation 2003/94/EC [90]. Serum-free culture protocols have been studied with the clear purpose to avoid these problems. During the last decade many serum-free based culture protocols, including some of our own group, have demonstrated not only a good viability of hDPSCs under the absence of FBS, but also a better preservation of their stem cell characteristics [80,81,91]. The main drawback is that growth rates of hDPSCs in serum-free media are usually lower than in the presence of FBS [80], which can constitute a limitation depending on the intended application. At the present time there is nothing matching FBS as a culture supplement to promote in vitro hDPSC proliferation and expansion, even though quite acceptable growth rates of hDPSC cultures can be obtained in serum free-media with the presence of growth factors like epidermal growth factor (EGF) or basic fibroblast growth factor (bFGF) [80,81,92]. However, irrespective of the presence or absence of fetal serum, it is never desirable to induce an excessive ex vivo expansion of hDPSC cultures, because of the risk to induce telomere attrition and its intrinsically associated cellular senescence [93].

When hDPSCs are cultured in serum-free media they tend to form floating or poorly adherent spheroids, in contrast to the clear plastic-adherent phenotype they show in the presence of FBS [80,91]. These spheroids, also termed pulp dentospheres, present some striking shared characteristics with brain neurospheres (Figure 3A) [80]; first, they show a high expression of NSC markers like Nestin and Glial Fibrillary Acidic Protein (GFAP); second, they can be maintained and grow in the same media of NSCs; third, upon switching to neural cell differentiation media, they generate neuron and glial marker-expressing cells in similar proportions to brain neurospheres. The formation of pulp dentospheres in serum free media is amply regarded as a condition which favors the generation of neural lineage cells from hDPSCs [58,92,94]. However, there exists a lot of variability in the literature with regard to the exact media composition to generate hDPSCs dentospheres. This lack of consensus can lead to diverse differentiation outcomes and/or lack of reproducibility.

Many of the reported hDPSC neural differentiation protocols adopt a two-stage culture strategy with two consecutive phases of expansion in floating dentospheres in serum free media containing EGF and bFGF, and a subsequent differentiation stage in adherent conditions, which takes place over substrates coated with adhesion proteins like laminin (Figure 3B) [92,95]. Some studies have also reported a direct neuronal-like differentiation of hDPSCs in adherent conditions, thus bypassing the need for dentosphere generation [96]. In order to increase reproducibility and improve the systematic comparison of neural differentiation efficiency from hDPSCs and NSCs, established cell culture protocols to grow NSCs could be tested on hDPSCs as well. We often find that hDPSCs are able to thrive in the same culture media that are currently used for the growth and differentiation of NSCs. There exist some commercial culture media (e.g., Neurocult™; Stem Cell Technologies Vancouver, BC, Canada) which are compatible with both hNSC and hDPSC growth [80]. The main drawback of these commercial media is that they often contain supplements of undisclosed composition, which make difficult to assess the molecular mechanisms of hDPSC neural differentiation because of the potential interaction of the experimental manipulations with those secret supplement media components.

## 4. Results of Neuronal and Glial Differentiation

DPSCs are derived from the neural crest during development, and one of their in vivo functions is to renew Schwann cell populations of the abundant nerve fibers innervating the dental pulp [79]. Neurovascular bundles of the dental pulp are extremely rich in peripheral nerve fibers, many of which are myelinated. Owing to the intimate relationship of hDPSCs with the peripheral nervous system, it is not surprising that they can in vitro differentiate to functional Schwann cells. A high proportion of hDPSCs exposed to neural differentiation media co-express p75NTR with S-100β [71], which constitutes the distinctive molecular marker profile of Schwann cells [97]. In vitro hDPSC-derived Schwann cells have been shown to help guide axonal extension and myelination [98,99]. These differentiation characteristics make hDPSCs very attractive candidates for the treatment of acute nerve crush injuries, as it has been extensively demonstrated in animal models [99,100]. Moreover, in experimental animal models of cell therapy for Spinal Cord Injury (SCI), many of the in situ grafted hDPSCs were shown to differentiate to myelinating cells, which correlated with functional recovery [101]. It is unclear whether hDPSCs or their derived Schwann cells could transdifferentiate to oligodendrocytes, when transplanted from their niche environment of the Peripheral Nervous System (PNS) to the much more strictly regulated environment of the brain and spinal cord. The in situ differentiation of hDPSCs to functional oligodendrocytes within the Central Nervous System (CNS) represents an attractive possibility that requires more investigation; however, the potential benefits of hDPSC grafts for SCI go far beyond the replacement of lost myelinating cells. Another very interesting characteristic of hDPSCs is their high expression levels of neurotrophic factors [80,100,102] which would not only enhance the survival of both resident neuronal and glial cells, but also participate in neurite chemoattraction to facilitate reinnervation.

Despite being a matter that has received considerable more attention than glial differentiation, the neuronal differentiation from hDPSCs is still subjected to controversy. Over the last years, many research papers have claimed the generation of neuronal marker-expressing cells from hDPSC cultures [58,85,86,92]. However, these reports should be examined with caution, because hDPSCs are long-known to naturally express mature neuronal markers, even in the absence of neurogenic stimuli. Thus, any relevance to be attributed to putative changes in the expression of a given neuronal marker by hDPSCs depends critically on the choice of that marker, and not all of them are appropriate for that matter (see some illustrative examples below). When the right markers are chosen, a case can be made for a hDPSC differentiation to neuron-like or neuronal-lineage related cells. However, if one wishes to go further in the claims of genuine neuronal differentiation, then accessory experiments should be included, such as an electrophysiological characterization of those putative hDPSC-derived neurons. It is precisely on this qualitative step where we have much less conclusive evidence, and many more questions. Early experiments by Arthur et al. showed that hDPSC-derived neuronal-lineage cells presented TTX-sensitive voltage-dependent Na^+^ currents [86] but they failed to show a single Action Potential (AP) on these cells. Later on, Gervois et al. managed to generate AP-like depolarizations in neurodifferentiated hDPSC-derived cells. Those fast depolarizations replicated the rising phase of the AP, but importantly, they showed no apparent recovery to the baseline potential, and no more than one of them could be induced at a time [92]. These findings were later corroborated by Li et al. [95]. In both cases very large current injections had to be applied, in the order of 100–300 pA for 1–2 s, which is far above what is usually required to induce APs in cultured neurons in vitro [103]. In contrast, when primary mNSCs differentiate into mature neurons, they can generate APs which show all the expected phases of fast depolarization, repolarization and subsequent recovery to baseline potential after a transient hyperpolarization, and these APs could be induced with short current injections of 1.5 nA for just 0.3–0.7 ms [104].

These evidences illustrate that generating bona fide neurons from hDPSCs could be a more complicated issue than expected. Besides the very different electrophysiological profile of neuronal APs, in vitro cultured neurons can also generate bursts of consecutive APs when they are subjected to depolarization [105]. The fact that no more than one AP-like depolarization could be induced in hDPSC-derived cells also suggests that the neuronal differentiation protocols tested so far were somehow incomplete, and failed to induce the full array of characteristics of genuine neurons on hDPSCs. Finally, to date there is no demonstration that hDPSC-derived neuron-like cells can establish functional synaptic contacts with neurons in co-culture. Some promising results have been shown such as the formation of cell-to-cell contacts containing synaptic proteins [106], but definite ultrastructural and electrophysiological evidence of synaptic coupling between neurons and hDPSCs is still lacking. In contrast, neurons obtained from NSCs, show very long and developed dendritic trees and axons, and an ultrastructural pattern of synaptic maturation comparable to in vivo synaptogenesis, including the appearance of pre-synaptic vesicles and post-synaptic densities by Transmission Electron Microscopy (TEM) [107].

## 5. Neuronal-Like Differentiation from hDPSCs: Problems and Confusions

One important problem when assessing neuronal differentiation is that non-differentiated hDPSCs already express many of the molecular markers that are traditionally used for identifying mature neurons. This is particularly true for some cytoskeleton-associated proteins like β-3 tubulin (Tuj1). Other cytoskeletal proteins that are also abundantly expressed by hDPSCs are the intermediate filaments Vimentin, Nestin and GFAP, which are commonly used as markers for NSCs [108]. Thus, hDPSCs combine the expression of all these markers, both from mature neurons and immature NSCs. Therefore, if one wished to assess the neuronal differentiation effect of a given experimental treatment or manipulation, logic would dictate that the expression of immature NSC markers would downregulate, and the expression of mature neuronal markers would concomitantly upregulate. However, still quite often these two different groups of markers are found to be indistinctly mixed in the literature as an evidence of neuronal differentiation (e.g., a simultaneous rise of Nestin, Sox2 and MAP2 [109], or Nestin, β-3 tubulin and GFAP [110]). Even more problematic can be the use of non-specific labeling methods like Nissl stain [111,112], or the confusion that a mere rise in the expression of a particular neuronal marker (that is already substantially expressed by control hDPSCs) is taken as indicative of neuronal differentiation. Take for instance the case of the mature neuronal marker β-3 tubulin (Figure 4A). The expression of β-3 tubulin by hDPSCs is already remarkably high, as assessed by different techniques like immunofluorescence (IF), western blot (WB), flow cytometry and quantitative polymerase chain reaction after retrotranscription (qPCR). Is it likely that an additional rise in β-3 tubulin expression after a given treatment indicates a neuronal differentiation, when practically 100% of control hDPSCs already express it? Then what about cellular morphology? Has it changed at all after the treatment, with respect to control hDPSCs? Do differentiated cells show very thin and long processes, like dendrites and axons, or rather the typical lamellipodia of fibroblast-like cells? Those are the type of questions that should be considered when critically assessing the obtained neurodifferentiation results.

The expression of neural markers by hDPSCs is unavoidable, but the impact this has in the assessment of the neurodifferentiation protocols can be partially offset by choosing some neuronal markers that are not naturally expressed by hDPSCs in control basal conditions. In our hands, two of the neuronal markers that gave the best results were Doublecortin (DCX) and NeuN, which were completely absent in control non-differentiated hDPSCs, but whose expression rose sharply after neural induction with Neurocult™ differentiation media, as shown by IF and qPCR (Figure 4B,C). With this optimal choice of markers, we obtained very similar rates of neuronal marker-expressing cells when comparing hDPSCs with murine NSCs grown in parallel with similar Neurocult™ differentiation media [80]. However, we were well aware that the cellular morphologies we obtained from hDPSC cultures were not always consistent with that of genuine neurons. Some of the NeuN/DCX positive cells derived from hDPSCs showed big lamellipodia and no particularly long and thin cellular processes after 1 week of differentiation [80]. As a comparison, the morphology of NeuN/DCX-expressing neuronal cells obtained from NSCs during the first week of differentiation was featured by a typical bipolar/multipolar shape with characteristically thin and elongated neurites [80,107]. Later on, at around 14 days of differentiation these NSC-derived cells usually become fully mature neurons showing extremely long branching dendrites and axons, and synapses identified by ultrastructural TEM analysis [107]. That has never been described yet for hDPSC-derived neuronal-like cells. Certainly, the results obtained by the research community and the future prospects are very promising and it is possible that one day soon fully functional neurons might be derived from hDPSCs in the absence of induced genetic modifications, but with the evidence available so far, we can only recommend humility and caution in order not to fall into an exaggeration of expectations. Protocols must be refined and results evaluated more rigorously to affirm categorically a neuronal differentiation from hDPSCs.

## 6. Neuroprotective and Immunomodulatory Capacity. Extracellular Vesicles

DPSCs are well-known to secrete neuroprotective growth factors such as nerve growth factor (NGF), brain-derived neurotrophic factor (BDNF), glial cell line derived neurotrophic factor (GDNF), neurotrophin 3 (NT-3), vascular endothelial growth factor (VEGF) and platelet-derived growth factor (PDGF) [102,113]. We described the presence of BDNF and NT-3 specific receptors in hDPSCs and their importance for their neural differentiation [71]. Both in vivo and in vitro studies demonstrated a higher expression of neuroprotective growth factors by hDPSCs, compared with other MSC types [114]. These growth factors are able to reduce the neurodegeneration in the early stages of neural apoptosis and sensory neuron survival [115]. Furthermore, some of them may promote axon regeneration and neurite outgrowth, even in spite of the presence of axonal growth inhibitors in vivo [102,116]. In ischemic injury models, DPSCs provide both direct and indirect cytoprotection [117]. For all these reasons, hDPSCs are regarded as a very promising therapeutic tool to restore neural function after trauma or disease. Not only these cells secrete high amounts of neuroprotective factors, but they are also particularly resistant to hypoxic/ischemic conditions [76], giving them a better chance of taking root and regenerating largely degraded areas of the brain with a compromised vasculature.

In the event of trauma or during the course of neurodegenerative diseases, cells of the immune system have been proposed to modulate the course of pathogenesis by their pro-inflammatory signals [118,119]. Whether intrinsic to the brain (microglia) or peripheral (myeloid cells), the activity of the immune system can generate a pro-inflammatory environment, causing a worsening of the lesion [119,120]. Thus, it is important to target inflammation and prevent the infiltration of immune system cells to preserve brain homeostasis and neural integrity and functionality. One method to assess the global levels of inflammation or the activation of the immune system consists of measuring the levels of soluble urokinase-type plasminogen activator receptor (suPAR) in blood [121,122]. suPAR is a protein that plays an essential function in leukocytes and endothelial homeostasis, and it can be detected even in patients with very mild periodontitis to severe cardiovascular disease or cancer, becoming an potential excellent biomarker for cell graft tolerance [121,123,124].

Interestingly, hDPSCs are also known to secrete strong immunomodulatory and anti-inflammatory cytokines such as Interleukin-8 (IL-8), Interleukin-6 (IL-6), Transforming Growth Factor Beta (TGF-β), Hepatocyte Growth Factor (HGF) and Indoleamine 2,3-dioxygenase (IDO) [125,126,127]. TGF-β, HGF and IDO are able to suppress both the activation of T cells, the proliferation of peripheral blood mononuclear cells, and even allogeneic immune responses [128,129]. In addition, IL-8 helps preserving axon integrity in SCI [130]. The co-culture of hDPSCs and T cells resulted in human leukocyte antigen-G, vascular adhesion molecule-1, intracellular adhesion molecule-1, IL-6, TGF-β, HGF, and IL-10 secretion. Moreover, pro-inflammatory IL-2, IL-6 receptor, IL-12, IL-17A and Tumor Necrosis Factor-α (TNF-α) cytokines were downregulated [131], and this induced a 90% reduction on the proliferation rate of T cells [132]. Therefore, hDPSCs as well as other MSCs possess strong immune-suppressive properties which make them useful to control brain inflammation.

One of the mechanisms of action of both MSCs and DPSCs to spread their immunomodulatory and anti-inflammatory signals is through their secreted Extracellular Vesicles (EVs) [133,134,135,136,137,138]. EVs are small lipid-bilayer membrane vesicles, secreted to the extracellular environment, which can carry different intraluminal loads such as proteins, lipids and nucleic acids [139]. EVs have been conserved throughout evolution from bacteria to humans [140]. Thus, the presence of stem cells and/or the administration of their EVs can exert a protective effect, by reducing inflammatory signals [141]. This has an enormous potential in regenerative medicine [142], to treat nervous system diseases [143] and even neurocognitive disorders [144]. However, EVs are not only released from DPSCs or MSCs [145,146] but from many other cells including endothelia, neurons, astrocytes, microglia and oligodendrocytes [147] constituting a widespread mode of cell communication in health and disease [148,149].

## 7. DPSCs in Combination with Scaffold Materials for Neuroregeneration

Although stem cells are widely used for nerve tissue regeneration, most of the preclinical studies demonstrated a poor functional integration into the host neural circuitry of the differentiated neuronal-like cells [150]. The combination of biomaterials with DPSCs represents a promising approach to enhance cell engraftment. Most of the in vitro research has been conducted with polymers from synthetic or natural origin, including polysaccharides such as chitosan, or proteins such as collagen [58,151].

Regarding natural polymers, chitosan scaffolds combined with hDPSCs are some of the most promising materials. The combination of chitosan 3D porous scaffolds with bFGF enhances the expression of neural markers in hDPSCs by the activation of the extracellular signal-regulated kinase (ERK) pathway in vitro [152]. Moreover, in vivo grafts of hDPSC/chitosan improved locomotor disability in animal models of SCI, by the secretion of BDNF, GDNF and NT-3, reducing the accumulation of active-caspase 3, and impairing axonal loss and degradation compared to the non-grafted animals. Interestingly, hDPSCs cultured in combination with chitosan were shown to activate Wnt/β-catenin signaling [153]. Wnt/β-catenin activation is known to promote the stemness and neural crest attributes of hDPSCs [73,74,75], which might explain a better neural differentiation capacity in these conditions.

Heparin-Poloxamer (HP), a thermosensitive material that achieves a hydrogel structure at body temperature [154], improves the functional locomotor recovery in SCI models in vivo when combining it with hDPSCs and bFGF. The HP-hDPSC grafts upregulate the antiapoptotic protooncogene B-cell lymphoma 2 (Bcl-2) and reduce the amount of the BCL-2 associated X apoptosis regulator (Bax) as well as other apoptotic markers such as cleaved Caspase-3, compared to control non-grafted animals [155]. In a related study, the combined graft of hDPSCs with thermosensitive heparin hydrogels containing bFGF was shown to efficiently reduce pro-inflammatory cytokine release in murine SCI [156].

Collagen is a fibrous protein of the extracellular matrix, which is also considered as a promising scaffold for neuroregeneration [157]. A subpopulation of hDPSCs positive for STRO-1, c-kit and CD34 promoted axonal growth, remyelination and peripheral nerve regeneration when they were transplanted with a collagen hydrogel in an in vivo sciatic nerve lesion, by the secretion of neurotrophic factors BDNF, NT-3 and NGF [158]. Interestingly, the grafting of collagen with Schwann-cell differentiated hDPSCs promoted the formation of new vascular tubes and axons, thanks to the secretion of neuroprotective factors like vascular endothelial growth factor A (VEGF-A), reflecting the perivascular origin of these cells [159]. In other studies, hDPSCs seeded inside a silicone and collagen gel tube were shown to generate a Schwann-cell rich tissue that contained blood vessels, secreted neurotrophic factors [160] and improved the electrophysiological and functional recovery of the facial nerve [161].

Synthetic polyesters like poly—ε caprolactone (PCL) or poly—lactide—co—glycolide (PLGA) tubes filled with hDPSCs can also improve axonal and facial nerve regeneration, by providing a biocompatible source of neurotrophic factors [162]. Some groups also studied the in vitro capabilities of other alternative synthetic materials like graphene-derivatives (e.g., graphene oxide (GO) or reduced graphene oxide (rGO)). These materials were recently postulated as a promising tool for NSC differentiation thanks to their biocompatibility, intrinsic electrical properties, and neurodifferentiation inductive capabilities (139). The absence of cytotoxic effects of GO [163], and the induction of neuronal cell lineage differentiation have been described on hDPSCs seeded on rGO-PCL hybrid electrospun nanofibers [164].

## 8. Perspectives on the Applications of hDPSCs in Neuroregenerative Therapies

hDPSCs are very attractive cell source to use in nerve tissue regeneration therapies due to their better neuronal and glial differentiation capacity, compared with that of other more conventional sources of adult multipotent stem cells like MSCs. hDPSCs may also help create an appropriate microenvironment to attract neurite outgrowth and reinnervation on very diverse types of neural lesions [106], even to treat retinal degeneration [165]. In murine models of neurodegenerative Parkinson’s disease, the intrathecal graft of hDPSCs ameliorated behavioral deficits and dopaminergic (DA) neuron loss, by upregulating anti-inflammatory cytokines IL2, IL4, and TNF-β and reducing IL-1α, IL- 1β, IL6, IL8, and TNF-α pro-inflammatory ones [166]. Using 1-methyl-4-phenyl-1,2,3,6-tetrahydropyridine (MPTP) and 6-hydroxydopamine (6-OHDA) PD models, hDPSCs demonstrated a neuroprotective activity over DA neurons [167,168,169,170]. The neurotrophic and anti-inflammatory potential of intracranial hDPSCs grafts also ameliorated striatal atrophy in a chemical rat model of Huntington’s disease [171]. In Alzheimer’s disease, a reduction of amyloid beta (Aβ) peptide-induced cytotoxicity and cellular apoptosis was linked to a secretion of neuroprotective factors by hDPSCs [172,173].

The ability of hDPSCs or their secreted EVs to induce vasculogenesis and angiogenesis [80,81,146,174] is essential in CNS regeneration, to provide nutrients and oxygen to the injured brain or spinal cord tissue. hDPSCs have a huge potential for the reestablishment of an injured CNS vasculature. Our research group showed that an intrahippocampal injection of hDPSCs generated fully developed blood vessels containing perfectly aligned endothelial cells, basement membranes and pericytes after a post-graft period of one month in the rodent brain [80]. Chances are high that hDPSCs could take root even on very destroyed or degraded areas of the CNS, contributing to their neovascularization. These cells are extraordinarily resistant to ischemia [76] and together with embryonic precursor cells they have been tested in animal models of ischemic disease [175]. Moreover, hDPSCs were found to improve post-stroke recovery in animal models of middle cerebral artery occlusion (MCAO) [176,177]. For all these reasons, the first clinical trials using hDPSCs against ischemic stroke are already on their way [178,179]. However, despite the recent progress on clinical applications, clinical trials with hDPSCs for neural pathologies are still very limited, especially in comparison with other types of MSCs [180]. In contrast to all the basic research that has been carried out with hDPSCs, the clinical translation of neuroregenerative cell therapies based on hDPSCs still seems to be relatively small, with the notable exceptions of ongoing clinical trials for stroke and Huntington’s disease [181].

## 9. Conclusions and Future Directions

In our opinion, hDPSCs present particularly attractive characteristics to be exploited in neuroregenerative cell therapies, such as their high capacity for neural differentiation, their vasculogenic, neurotrophic, and immunomodulatory properties, and their easy applicability to autologous therapy. On the downside, the amount of tissue that can be isolated from the human dental pulp to extract hDPSCs is relatively small per single biopsy, precluding the collection of so many stem cells as in the case of other more traditional sources. Despite these limitations, hDPSCs can be cultured with acceptable growth rates in completely serum free media [80,81,91], and it is also possible to devise strategies to better preserve the neural crest phenotype and stemness of hDPSC cultures even after a fast initial phase of cellular expansion in the presence of fetal animal serum. This can be accomplished for instance by a transient in vitro stimulation of hDPSCs with Wnt signaling activators [73] and neurotrophins [71]. For all these reasons, it is very likely that the number of preliminary clinical trials using hDPSCs against neural diseases will see a surge in the near future, as it is now being the case for the current COVID-19 pandemic [182,183].

## Figures and Tables

**Figure 1 ijms-22-03546-f001:**
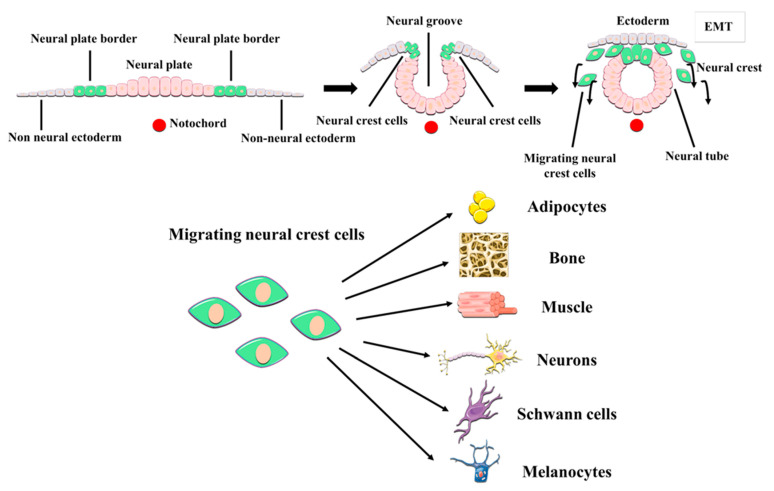
Embryonic origin and multilineage differentiation of human dental pulp stem cells (hDPSCs). hDPSCs derive from neural crest stem cells that generate craniomaxillofacial tissues, including the dental pulp. During development, neural crest cells undergo an epithelial-mesenchymal transition (EMT) and migrate out of the neural tube, to give rise to both mesenchymal and non-mesenchymal cell lineages of the oral cavity, like the neurons and glial cells of the craniofacial PNS. hDPSCs show many neural crest characteristics such as their expression of neural crest markers, and a higher differentiation potential to neural cell lineages than other MSCs.

**Figure 2 ijms-22-03546-f002:**
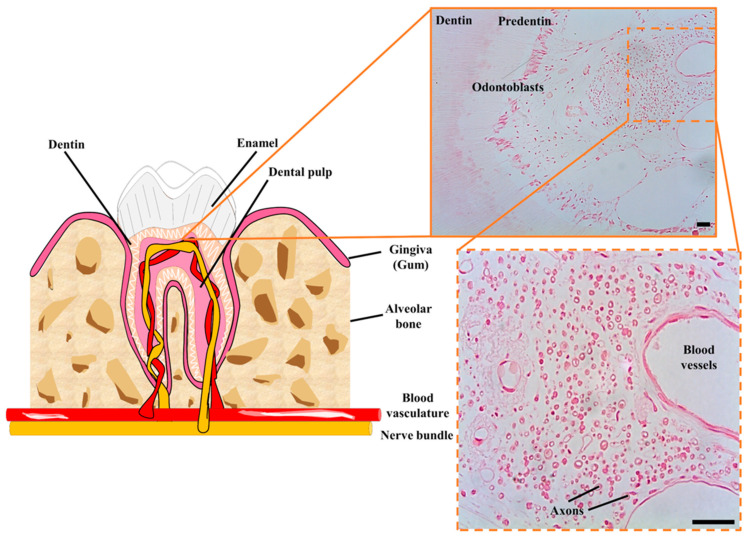
Cellular niche of hDPSCs in postnatal teeth. hDPSCs are harbored in neurovascular bundles of the dental pulp of mature teeth, containing a high concentration of nerve fibers and blood vessels. These neurovascular niches contain many myelinated axons (shown in cross-section) and a higher cellular density than in the rest of the dental pulp tissue. Scale bars: 50 µm.

**Figure 3 ijms-22-03546-f003:**
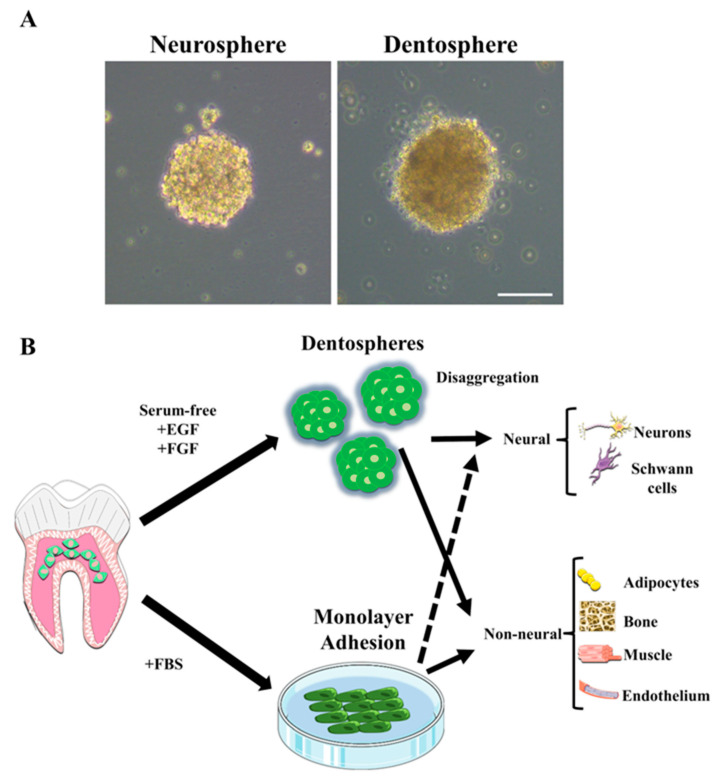
Culture protocols to induce neural differentiation of hDPSCs. (**A**) When grown in the same serum-free media, hDPSCs form pulp dentospheres which are morphologically and functionally similar to brain neurospheres. (**B**) Like neurospheres, dentospheres can be disaggregated to generate neuronal and glial marker expressing cells in the presence of neural induction media. The same hDPSCs in the presence of fetal serum (FBS) differentiate preferentially to non-neural cell lineages. However, it is also possible to generate non-neural cells from pulp dentospheres in serum-free media, and also to induce a neural-like differentiation from hDPSCs when these are switched from serum-containing to serum-free neural induction media.

**Figure 4 ijms-22-03546-f004:**
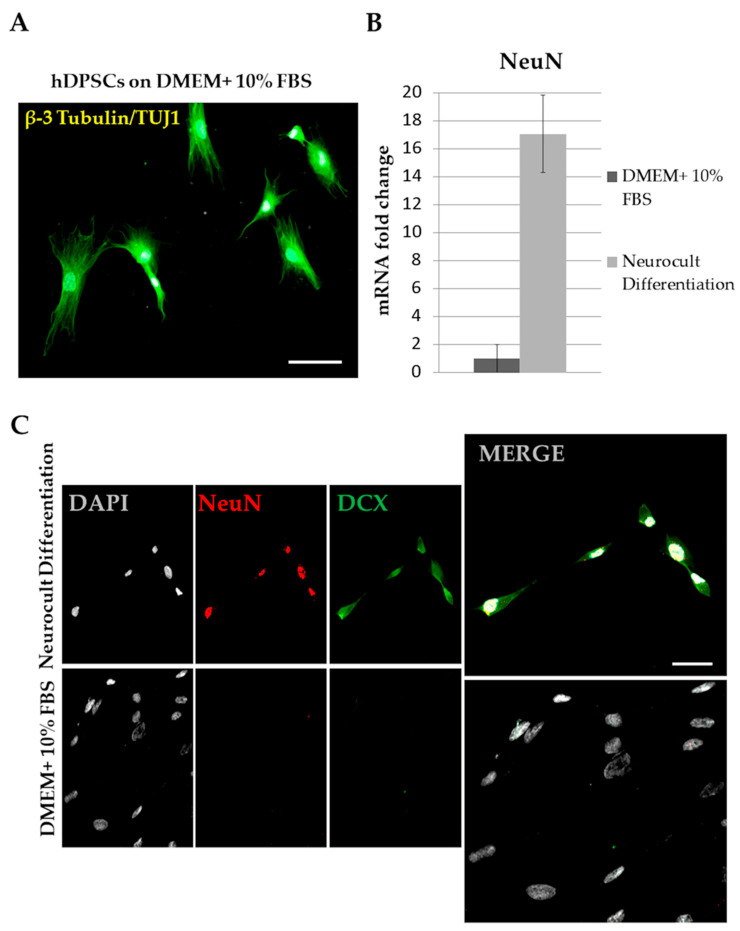
Choosing markers to assess a neuronal-like differentiation of hDPSCs. (**A**) hDPSCs in control non-neural inductive culture conditions still express some mature neuronal markers at high levels, as it is the case of β-3 tubulin/Tuj 1. (**B**) Some other neuronal markers like NeuN are not expressed at all in control conditions, as assessed by qPCR (error bar overlaps with zero Y-axis value on the graph; n = 3). However, when switching hDPSCs from control media to neural differentiation media, NeuN expression rises sharply, by more than 15× (n = 3). (**C**) Comparison of the expression of the neuronal markers DCX and NeuN, as assessed by IF, for hDPSCs grown in control (DMEM+ 10% FBS) and Neural induction (Neurocult™ differentiation) media. Note the difference of expression of DCX/NeuN between both culture conditions, and the predominantly cytoplasmic staining of DCX with respect to the nuclear staining of NeuN.

## Data Availability

Not applicable.

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
