# Peer review of "Advances and Perspectives in Dental Pulp Stem Cell Based Neuroregeneration Therapies"

_ijms, 2021, doi:10.3390/ijms22073546_

Round 1

Reviewer 1 Report

This manuscript reviewed the role of DSPCs serving as potential stem cells for neural regeneration. Neural stem cells (NSCs) give rise to both neurons and glial cells. Because of the limitations of endogenous NSCs, MSCs were extensively tested for neural regeneration and obtained promising results. The authors provided a fine review about multilineage differentiation of DSPCs and discussion about potential neuroregeneration therapies. However, this reviewer has the following comments on this manuscript.

It is very interesting that both NSCs and DSPCs can form neurospheres and pulp dental spheres. It would be helpful to compare the difference and similarity in neural regeneration between NSCs and DSPCs besides neurosphere formation. For example, can NSCs differentiate into neurons which were able to induce action potential (AP)?

Line 376 stated:"to date there is no demonstration that hDPSC-derived neuron-like cells can establish functional synaptic contacts with neurons in co-culture".

Reviewer's comment: Can NSC-derived-like cells establish functional synaptic contacts with neurons in co-culture?

Line 431 stated: "well aware that the cellular morphologies we obtained from hDPSC cultures were not always consistent with that of genuine neurons>"

Reviewer's comment: Was the cellular morphologies from NSC cultures consistent with that of genuine neurons?

It would be helpful to include the potential role of induced pluripotent stem cells in neuroregeneration therapies in the manuscript.

The scarce numbers of endogenous NSCs are available for neural tissue regeneration. Thus MSCs have been used for neural regeneration. DPSCs are categorized as MSCs. Furthermore, DPSCs derived from neural crest cells, were able to differentiate to neural cells and glial cells. DSPCs were a good candidate for neuroregeneration. However, the amount of extracted DPSCs was small per single biopsy. Did authors speculate any strategy to overcome the low DPSC cell supply issue?

A minor comment

on line 179-180, reference [69]: "in contrast to the surrounding loose connective tissue of the rest of the dental pulp [69]"

Reviewer comment: Please list the page for that statement.

Author Response

Reviewer 1

This manuscript reviewed the role of DSPCs serving as potential stem cells for neural regeneration. Neural stem cells (NSCs) give rise to both neurons and glial cells. Because of the limitations of endogenous NSCs, MSCs were extensively tested for neural regeneration and obtained promising results. The authors provided a fine review about multilineage differentiation of DSPCs and discussion about potential neuroregeneration therapies. However, this reviewer has the following comments on this manuscript.

 It is very interesting that both NSCs and DSPCs can form neurospheres and pulp dental spheres. It would be helpful to compare the difference and similarity in neural regeneration between NSCs and DSPCs besides neurosphere formation.

We thank the reviewer for this comment, as we believe it is a very meaningful suggestion. Thus, we have deepened in the comparison of the characteristics of NSCs and DPSC-generated neuronal-like cells throughout this new manuscript version, regarding AP generation, synapse generation, neuronal markers and cellular morphology, etc.

For example, can NSCs differentiate into neurons which were able to induce action potential (AP)?

ANSWER TO REVIEWER: Yes, NSCs can differentiate into glial cells and neurons which show neuronal APs with all the expected depolarizing, repolarizing and hyperpolarizing phases, and a final recovery to baseline potential (as shown in Gritti et al. J Neurosci 2006. https://pubmed.ncbi.nlm.nih.gov/8558238/ ).

We added the following sentence to the part of the manuscript where we compared APs generated from neuronal-like cells generated from hDPSC and NSCs (page 12):

“In contrast, when primary mNSCs differentiate into mature neurons, they can generate APs which show all the expected phases of fast depolarization, repolarization and subsequent recovery to baseline potential after a transient hyperpolarization, and these APs could be induced with short current injections of 1.5 nA for just 0.3-0.7 msec [101]”.

REFERENCES:

A Gritti, E A Parati, L Cova, P Frolichsthal, R Galli, E Wanke, L Faravelli, D J Morassutti, F Roisen, D D Nickel, A L Vescovi. Multipotential stem cells from the adult mouse brain proliferate and self-renew in response to basic fibroblast growth factor. J Neurosci 1996 Feb 1;16(3):1091-100.

Line 376 stated:"to date there is no demonstration that hDPSC-derived neuron-like cells can establish functional synaptic contacts with neurons in co-culture".

Reviewer's comment: Can NSC-derived-like cells establish functional synaptic contacts with neurons in co-culture?

ANSWER TO REVIEWER: Yes, NSCs are able to establish synaptic contacts in vitro. Indeed there is a work entitled “Maturation of Synaptic Contacts in Differentiating Neural Stem Cells” from Liebau et al. (Stem Cells 2007 https://doi.org/10.1634/stemcells.2006-0823) describing it.

We added to the manuscript the following paragraph when talking about the establishment of synapses, comparing results of hDPSCs and NSCs (page 12):

“In contrast, neurons obtained from NSCs, show very long and developed dendritic trees and axons, and an ultrastructural pattern of synaptic maturation comparable to in vivo synaptogenesis, including the appearance of pre-synaptic vesicles and post-synaptic densities by TEM [104].”

REFERENCES:

Liebau, S.; Vaida, B.; Storch, A.; Boeckers, T.M. Maturation of Synaptic Contacts in Differentiating Neural Stem Cells. Stem Cells 2007, 25, 1720–1729, doi:10.1634/stemcells.2006-0823.

Line 431 stated: "well aware that the cellular morphologies we obtained from hDPSC cultures were not always consistent with that of genuine neurons>"

Reviewer's comment: Was the cellular morphologies from NSC cultures consistent with that of genuine neurons?

ANSWER TO REVIEWER: Yes. In the work entitled “Maturation of Synaptic Contacts in Differentiating Neural Stem Cells” from Liebau et al. (Stem Cells 2007 https://doi.org/10.1634/stemcells.2006-0823) they clearly describe axonal and dendritic compartments of the NSCs differentiated neurons from 14 days of differentiation (DIV) onwards, and differentiated NSCs that perfectly mirrored hippocampal pyramidal neuron morphology in culture at 23DIV (page 1726). In our own work, we were able to partially reproduce the results of Liebau et al. regarding cellular morphology of NSC-derived cells during the first 7 days of neuronal differentiation (Luzuriaga et al. 2019).

We have added the following sentences to the manuscript (pages 13-14; last-first paragraph):

“Some of the NeuN/DCX positive cells derived from hDPSCs showed big lamellipodia and no particularly long and thin cellular processes after 1 week of differentiation [79]. As a comparison, the morphology of NeuN/DCX-expressing neuronal cells obtained from NSCs during the first week of differentiation was featured by a typical bipolar/multipolar shape with characteristically thin and elongated neurites [79,104]. Later on, at around 14 days of differentiation these NSC-derived cells usually become fully mature neurons showing extremely long branching dendrites and axons, and synapses identified by ultrastructural TEM analysis [104]. That has never been described yet for hDPSC-derived neuronal-like cells.”

REFERENCES:

Liebau, S.; Vaida, B.; Storch, A.; Boeckers, T.M. Maturation of Synaptic Contacts in Differentiating Neural Stem Cells. Stem Cells 2007, 25, 1720–1729, doi:10.1634/stemcells.2006-0823. 

Luzuriaga, J.; Pastor-Alonso, O.; Encinas, J.M.; Unda, F.; Ibarretxe, G.; Pineda, J.R. Human Dental Pulp Stem Cells Grown in Neurogenic Media Differentiate Into Endothelial Cells and Promote Neovasculogenesis in the Mouse Brain. Front. Physiol. 2019, 10, 347

It would be helpful to include the potential role of induced pluripotent stem cells in neuroregeneration therapies in the manuscript.

ANSWER TO REVIEWER: We agree with this suggestion, and we included references about the high capability of embryonic and induced pluripotent stem cells to differentiate to neural cells, as well as some papers which thoroughly discuss the potential applications of IPSCs to neuroregenerative therapy:

We have added the following sentences to the manuscript (page 3; first paragraph):

“Stem cells from the spinal cord of 8-week fetuses  have been tested in clinical trials for chronic spinal cord injury [47,48]. However, it is unlikely that these strategies will ever reach a widespread implementation, due to the scarcity of embryo donors and the associated ethical issues. Induced pluripotent stem cells (IPSCs) have been proposed to overcome ethical concerns about the use of human embryos. IPSCs can be very efficently differentiated to neurons and glial cells [49–51] and they have been proposed as a promising alternative for cell therapy in brain and spinal cord injury [39], as a tool to screen genetic bases of neurological diseases [52] or even as an approach to correct alterations in chronic neurodegenerative diseases [53]. However, a better understanding is still required to regulate the generation of specific neuronal and glial populations in a balanced and coordinated manner. Furthermore, the increased risk of cancer related to the use of IPSCs is regarded as a major drawback for autologous personalized neuroregenerative therapy [49].”

REFERENCES:

Guo, X.; Johe, K.; Molnar, P.; Davis, H.; Hickman, J. Characterization of a Human Fetal Spinal Cord Stem Cell Line NSI-566RSC and Its Induction to Functional Motoneurons. J Tissue Eng Regen Med 2010, 4, 181–193, doi:10.1002/term.223.

Curtis, E.; Martin, J.R.; Gabel, B.; Sidhu, N.; Rzesiewicz, T.K.; Mandeville, R.; Van Gorp, S.; Leerink, M.; Tadokoro, T.; Marsala, S.; et al. A First-in-Human, Phase I Study of Neural Stem Cell Transplantation for Chronic Spinal Cord Injury. Cell Stem Cell 2018, 22, 941-950.e6, doi:10.1016/j.stem.2018.05.014.

Ford, E.; Pearlman, J.; Ruan, T.; Manion, J.; Waller, M.; Neely, G.G.; Caron, L. Human Pluripotent Stem Cells-Based Therapies for Neurodegenerative Diseases: Current Status and Challenges. Cells 2020, 9, doi:10.3390/cells9112517 

Hou, P.; Li, Y.; Zhang, X.; Liu, C.; Guan, J.; Li, H.; Zhao, T.; Ye, J.; Yang, W.; Liu, K.; et al. Pluripotent Stem Cells Induced from Mouse Somatic Cells by Small-Molecule Compounds. Science 2013, 341, 651–654, doi:10.1126/science.1239278. 

Sepehrimanesh, M.; Ding, B. Generation and Optimization of Highly Pure Motor Neurons from Human Induced Pluripotent Stem Cells via Lentiviral Delivery of Transcription Factors. Am. J. Physiol. Cell Physiol. 2020, 319, C771–C780, doi:10.1152/ajpcell.00279.2020

Klapper, S.D.; Garg, P.; Dagar, S.; Lenk, K.; Gottmann, K.; Nieweg, K. Astrocyte Lineage Cells Are Essential for Functional Neuronal Differentiation and Synapse Maturation in Human IPSC-Derived Neural Networks. Glia 2019, 67, 1893–1909, doi:https://doi.org/10.1002/glia.23666.

Toft, M. Advances in Genetic Diagnosis of Neurological Disorders. Acta Neurol. Scand. Suppl. 2014, 20–25, doi:10.1111/ane.12232

Li, Y.; Shen, P.-P.; Wang, B. Induced Pluripotent Stem Cell Technology for Spinal Cord Injury: A Promising Alternative Therapy. Neural Regen Res 2021, 16, 1500–1509, doi:10.4103/1673-5374.303013.

Toft, M. Advances in Genetic Diagnosis of Neurological Disorders. Acta Neurol Scand Suppl 2014, 20–25, doi:10.1111/ane.12232.

Fatima, A.; Gutiérrez-Garcia, R.; Vilchez, D. Induced Pluripotent Stem Cells from Huntington’s Disease Patients: A Promising Approach to Define and Correct Disease-Related Alterations. Neural Regen Res 2019, 14, 769–770, doi:10.4103/1673-5374.249223. 

The scarce numbers of endogenous NSCs are available for neural tissue regeneration. Thus MSCs have been used for neural regeneration. DPSCs are categorized as MSCs. Furthermore, DPSCs derived from neural crest cells, were able to differentiate to neural cells and glial cells. DSPCs were a good candidate for neuroregeneration. However, the amount of extracted DPSCs was small per single biopsy. Did authors speculate any strategy to overcome the low DPSC cell supply issue?

ANSWER TO REVIEWER: We agree with the reviewer that the small amount of cells obtained per single biopsy represents a setback for optimal cell therapy. However, a strategy to improve the number of cells is the pre-amplification or cell expansion in culture as it is currently done for skin transplants. One important aspect of cell culture is to avoid as much as possible the presence of animal serum. We and others have previously published protocols to culture DPSCs in a total absence of serum with very positive results (Luzuriaga J et al. Front Phys 2019. Luzuriaga J et al Biomedicines 2020). Moreover, it is possible to preserve the neurogenic capabilities of these cells even after an initial phase of cellular expansion in the presence of 10% FBS. Some culture modifications like the exposure to neurotrophins and to activators of the Wnt pathway have been reported to enhance the neural crest characteristics and stemness of hDPSCs (Luzuriaga et al. Cell Phys Biochem 2019, Uribe-Etxebarria et al. Eur. Cell. Mater. 2017).

We have added the following sentences to the manuscript (page 20; last paragraph):

“Despite these limitations, hDPSCs can be cultured with acceptable growth rates in completely serum free media[79,80,88], and it is also possible to devise strategies to better preserve the neural crest phenotype and stemness of hDPSC cultures, even after an initial phase of fast cellular expansion in the presence of fetal animal serum. This can be accomplished for instance by a transient in vitro stimulation with Wnt signaling activators [73] and neurotrophins [71].”

REFERENCES:

Luzuriaga, J.; Pastor-Alonso, O.; Encinas, J.M.; Unda, F.; Ibarretxe, G.; Pineda, J.R. Human Dental Pulp Stem Cells Grown in Neurogenic Media Differentiate Into Endothelial Cells and Promote Neovasculogenesis in the Mouse Brain. Front Physiol 2019, 10, 347, doi:10.3389/fphys.2019.00347.

Luzuriaga, J.; Irurzun, J.; Irastorza, I.; Unda, F.; Ibarretxe, G.; Pineda, J.R. Vasculogenesis from Human Dental Pulp Stem Cells Grown in Matrigel with Fully Defined Serum-Free Culture Media. Biomedicines 2020, 8, doi:10.3390/biomedicines8110483.

Luzuriaga, J.; Pineda, J.R.; Irastorza, I.; Uribe-Etxebarria, V.; García-Gallastegui, P.; Encinas, J.M.; Chamero, P.; Unda, F.; Ibarretxe, G. BDNF and NT3 Reprogram Human Ectomesenchymal Dental Pulp Stem Cells to Neurogenic and Gliogenic Neural Crest Progenitors Cultured in Serum-Free Medium. Cell. Physiol. Biochem. Int. J. Exp. Cell. Physiol. Biochem. Pharmacol. 2019, 52, 1361–1380

Uribe-Etxebarria, V.; Luzuriaga, J.; García-Gallastegui, P.; Agliano, A.; Unda, F.; Ibarretxe, G. Notch/Wnt Cross-Signalling Regulates Stemness of Dental Pulp Stem Cells through Expression of Neural Crest and Core Pluripotency Factors. Eur. Cell. Mater. 2017, 34, 249–270

 A minor comment

on line 179-180, reference [69]: "in contrast to the surrounding loose connective tissue of the rest of the dental pulp [69]"

Reviewer comment: Please list the page for that statement.

The exact page number is 423 (Nancy, A. Ten Cate’s Oral Histology - 9th Edition; Chapter 8: Dentin Pulp Complex; section name: “Undifferentiated ectomesenchymal cells”).

Reviewer 2 Report

In the manuscript entitled: “Advances and perspectives in dental pulp stem cell based neuroregeneration therapies” the authors investigated the knowledge about the embryonic origin and characteristics of their postnatal niche, as well as the current status of cell culture protocols to maximize their multilineage differentiation potential, highlighting some common issues when assessing neuronal differentiation fates of hDPSCs.

The author's found and discussed about recent progress on neuroprotective and immunomodulatory capacity of hDPSCs and their secreted extracellular vesicles, as well as their combination with scaffold materials to improve their functional integration on the injured central nervous system (CNS) and peripheral nervous system (PNS). Finally, we offer some perspectives on the current and possible future applications of hDPSCs in neuroregenerative cell therapies.

Major comments:

In general, the idea and innovation of this study, regards the analysis of dental pulp stem cell based is interesting, because the role of these cells in inflammatory diseases are validated, but further studies on this topic could be an innovative issue in this field could be open a creative matter of debate in literature by adding new information. Moreover, there are few reports in the literature that studied this exciting topic with this kind of study design.

The study was well conducted by the authors; However, there are some concerns to revise that are described below.

The introduction section and the manuscript resume the existing knowledge regarding the important factor linked with stem cells in dentistry.

However, as the importance of the topic, the reviewer strongly recommends, before a further re-evaluation of the manuscript, to update the literature through read, by must discuss and cites in the references with great attention all of those recent interesting articles, that helps the authors to better introduce and discuss the aim of the study in light of stem cells in periodontitis, mediators, oxidative stress and risk of systemic disease: 1) Isola G, Polizzi A, Alibrandi A, Williams RC, Leonardi R. Independent impact of periodontitis and cardiovascular disease on elevated soluble urokinase-type plasminogen activator receptor (suPAR) levels. J Periodontol. 2020 Oct 22. doi: 10.1002/JPER.20-0242.

The authors should be better specified at the end of the introduction section, the rationale of the study. In the discussion, should better clarify the role of stem cells, suPAR and TGF beta in the first stages following periodontal infection on subsequent healing.

The conclusion should be added with the main findings of the study and reinforce in light of the future directions.

In conclusion, I am sure that the authors are excellent clinicians who achieve very nice results with their adopted protocol. However, this study, in my view, does not in its current form satisfy a very high scientific requirement for publication in this journal and requests a revision before a further re-evaluation of the manuscript.

Minor Comments:

Introduction:

  • Please refer to major comments

Discussion

  • Please add a specific sentence that clarifies the results obtained in the first part of the discussion

Author Response

Reviewer 2

In the manuscript entitled: “Advances and perspectives in dental pulp stem cell based neuroregeneration therapies” the authors investigated the knowledge about the embryonic origin and characteristics of their postnatal niche, as well as the current status of cell culture protocols to maximize their multilineage differentiation potential, highlighting some common issues when assessing neuronal differentiation fates of hDPSCs.

The author's found and discussed about recent progress on neuroprotective and immunomodulatory capacity of hDPSCs and their secreted extracellular vesicles, as well as their combination with scaffold materials to improve their functional integration on the injured central nervous system (CNS) and peripheral nervous system (PNS). Finally, we offer some perspectives on the current and possible future applications of hDPSCs in neuroregenerative cell therapies.

 Major comments:

In general, the idea and innovation of this study, regards the analysis of dental pulp stem cell based is interesting, because the role of these cells in inflammatory diseases are validated, but further studies on this topic could be an innovative issue in this field could be open a creative matter of debate in literature by adding new information. Moreover, there are few reports in the literature that studied this exciting topic with this kind of study design.

We thank the reviewer for this positive assessment of the work. Our idea was to make an innovative review by comparing the characteristics of NSCs and DPSCs, as well as their neural cell derivatives. We think NSCs constitute the gold standard that all alternative sources of stem cells should compare with at the time of assessing their neuroregenerative potential. Interestingly, we found nothing of the sort on the existing scientific literature, which encouraged us to write this review article.

The study was well conducted by the authors; However, there are some concerns to revise that are described below.

The introduction section and the manuscript resume the existing knowledge regarding the important factor linked with stem cells in dentistry.

ANSWER TO REVIEWER: We must argue here that the main focus of this review was not to discuss the applications of hDPSCs in dentistry. There are already many excellent reviews on that topic. We wanted to do something completely different.

Thus, our main goal was to discuss the cellular and molecular aspects that are important to take into consideration for neuroregenerative cell therapies using hDPSCs, especially to treat pathologies of the Central Nervous System (CNS), by comparing hDPSCs with the endogenous type of stem cell of the CNS: the NSC. Of course, some aspects of hDPSC physiology (such as the production of immunomodulatory factors) can also be found in other reviews, but the focus we wanted to give to this manuscript was quite far away from dentistry.

However, as the importance of the topic, the reviewer strongly recommends, before a further re-evaluation of the manuscript, to update the literature through read, by must discuss and cites in the references with great attention all of those recent interesting articles, that helps the authors to better introduce and discuss the aim of the study in light of stem cells in periodontitis, mediators, oxidative stress and risk of systemic disease: 1) Isola G, Polizzi A, Alibrandi A, Williams RC, Leonardi R. Independent impact of periodontitis and cardiovascular disease on elevated soluble urokinase-type plasminogen activator receptor (suPAR) levels. J Periodontol. 2020 Oct 22. doi: 10.1002/JPER.20-0242.

ANSWER TO REVIEWER: We think that the proposed reference is quite interesting because it discusses suPAR as a very sensitive biomarker of inflammation and immune activation. This could serve to monitor the immune-tolerance of the grafted stem cells in human patients, and it could be of critical importance in the case of the CNS.

We have added the following sentences to the manuscript (page 17; first paragraph). The proposed reference Isola et al. 2020 is also included.

“One method to assess the global levels of inflammation or the activation of the immune system consists of measuring the levels of soluble urokinase-type plasminogen activator receptor (suPAR) in blood [119,120]. suPAR is a protein that plays an essential function in leukocytes and endothelial homeostasis, and it can be detected even in patients with very mild periodontitis to severe cardiovascular disease or cancer, becoming an potential excellent biomarker for cell graft tolerance [119,121,122].”

The authors should be better specified at the end of the introduction section, the rationale of the study. In the discussion, should better clarify the role of stem cells, suPAR and TGF beta in the first stages following periodontal infection on subsequent healing.

ANSWER TO REVIEWER: Actually, this is a review article and not an original research study. So the typical structure of Introduction, M&M, Results and Discussion of a conventional research report would not really apply here. Regarding the rationale/motivation of the review, we think it is quite well addressed when discussing the limitations of endogenous stem cells (NSCs) with neuroregeneration potential for cell therapy.

The conclusion should be added with the main findings of the study and reinforce in light of the future directions.

ANSWER TO REVIEWER: We have added a final section entitled: “9.Conclusions and Future directions”, where we outline what is possibly the main limitation of translational research with hDPSCs, namely, the relatively small size of collected material after pulp biopsy. We also suggest some alternative strategies to overcome this problem, with a view to the future.

In conclusion, I am sure that the authors are excellent clinicians who achieve very nice results with their adopted protocol. However, this study, in my view, does not in its current form satisfy a very high scientific requirement for publication in this journal and requests a revision before a further re-evaluation of the manuscript.

ANSWER TO REVIEWER: We thank the reviewer for these positive comments about our expertise. We are actually not clinicians, but basic researchers working in the Medicine Faculty and School of Engineering of the University of the Basque Country.

We have honestly tried to make the best review we could manage to cover a complex research topic with an innovative perspective. We hope the reviewer appreciates the efforts we made to improve the manuscript after this revision.

 Minor Comments:

 Introduction: Please refer to major comments

Discussion: Please add a specific sentence that clarifies the results obtained in the first part of the discussion

As already commented, there is no “Discussion” section as such in this manuscript.

Round 2

Reviewer 2 Report

The authors have well addressed to the reviewer's comment.

Author Response

Thank you for your feedback